# Ferroptosis: A Promising Therapeutic Target for Neonatal Hypoxic-Ischemic Brain Injury

**DOI:** 10.3390/ijms23137420

**Published:** 2022-07-04

**Authors:** Eric S. Peeples, Thiago C. Genaro-Mattos

**Affiliations:** 1Department of Pediatrics, University of Nebraska Medical Center, Omaha, NE 68198, USA; 2Children’s Hospital & Medical Center, Omaha, NE 68114, USA; 3Child Health Research Institute, Omaha, NE 68198, USA; 4Munroe-Meyer Institute for Genetics and Rehabilitation, University of Nebraska Medical Center, Omaha, NE 68198, USA

**Keywords:** phospholipid peroxidation, glutathione, GPX4, iron free radicals, neuroprotection, encephalopathy, asphyxia, FSP1

## Abstract

Ferroptosis is a type of programmed cell death caused by phospholipid peroxidation that has been implicated as a mechanism in several diseases resulting from ischemic-reperfusion injury. Most recently, ferroptosis has been identified as a possible key injury mechanism in neonatal hypoxic-ischemic brain injury (HIBI). This review summarizes the current literature regarding the different ferroptotic pathways, how they may be activated after neonatal HIBI, and which current or investigative interventions may attenuate ferroptotic cell death associated with neonatal HIBI.

## 1. Introduction

Neonatal hypoxic-ischemic encephalopathy—the clinical phenotype resulting from neonatal hypoxic-ischemic brain injury (HIBI)—is a devastating neurological injury following decreased oxygen and blood flow to the brain around the time of birth. Every year, nearly 1.2 million newborns worldwide suffer from encephalopathy, and despite treatment with therapeutic hypothermia, almost half of infants with moderate to severe HIBI die or suffer from significant developmental disability [1,2]. Since high morbidity and mortality occur after neonatal HIBI despite available therapy, novel supplemental therapies are needed.

Ferroptosis is a relatively recently described mechanism of cell death that is distinct from other types of cell death such as apoptosis, necrosis, necroptosis, etc. [3]. Ferroptosis was first described in 2012 by Dixon, et al. [4]. In their investigation of non-apoptotic mechanisms of cell death, they documented that cell death induced by erastin or RAS-selective lethal 3 (RSL3) lacked the characteristic features of apoptosis such as mitochondrial cytochrome c release and caspase activation. Instead, they demonstrated that the cell death mechanism that they termed ferroptosis was primarily characterized morphologically by iron-dependent accumulation of lipid reactive oxygen species and alterations in several ferroptosis-related genes such as iron response element binding protein 2 (IREB2), citrate synthase (CS), and acyl-CoA synthetase family member 2 (ACSF2).

The modulation of ferroptosis has been recognized as a potential target for novel therapies against numerous diseases [5]. Ferroptosis has been linked to the pathophysiology of multiple disorders of ischemia-reperfusion, including intestinal ischemia [6], myocardial infarction [7], and ischemic stroke [8]. Ferroptosis was recently identified as a key mediator of cortical mitochondrial damage [9] and hippocampal neuronal death [10] after neonatal HIBI. Despite the known involvement of ferroptosis in these conditions, the endogenous mechanisms remain poorly understood.

The timing of intervention is likely as important as the target, because neonatal HIBI is a triphasic injury [11]. The first phase initiates with the hypoxic-ischemic insult, the primary energy failure and hypoperfusion resulting in necrotic cell death in the minutes to hours after the restoration of normoxia and adequate perfusion. The secondary phase starts between 6 and 12 h after injury and continues until about 72 h after injury. It is characterized by deteriorating mitochondrial function and an acute inflammatory response, with apoptosis being the hallmark cell death process during this phase, though this is also the timing where ferroptotic changes have been observed (24–72 h [9,10]). After the secondary phase is the tertiary phase, which includes partial recovery and continued inflammation and gliosis [12]. When possible, this review will note the timing of the mechanisms and interventions studied to attempt to better understand the temporal changes in ferroptosis after neonatal HIBI.

Overall, this manuscript will review ferroptosis pathways and the role ferroptosis may play in HIBI-related cell death as well as potential therapeutic interventions to target neonatal HIBI. A recent review provided a general discussion of ferroptosis in neonatal brain injury (including preterm injury such as intraventricular hemorrhage) [13]; here, we will update the literature and focus on a more in-depth description of neonatal HIBI, including a comprehensive discussion of current investigative therapies that may target ferroptosis after neonatal HIBI.

## 2. Ferroptosis Pathways

Although oxygen-glucose deprivation has been the primary method of modeling hypoxia-ischemia in vitro, much like the clinical pathophysiology of HIBI, oxygen-glucose deprivation results in the activation of several parallel cell death mechanisms [14,15]. As such, the majority of the research described in this section will focus on pure ferroptosis models. These are usually generated by inducing ferroptosis through the inhibition of cystine transport (through the system Xc- pathway) by administering erastin or extracellular glutamate or the direct inhibition of glutathione peroxidase 4 (GPX4) with the chemical (1S,3R)-RSL3.

### 2.1. Phospholipid Peroxidation and Ferroptosis

Oxidative stress, lipid peroxidation, and cell death have long been implicated in many (patho)physiological conditions, including cancer, neurodegeneration, and ischemia-reperfusion injuries [16,17,18,19,20]. The specific association between lipid peroxidation and cell death was described in early studies linking the hepatotoxicity of carbon tetrachloride (CCl_4_) to lipid peroxidation [21]. Lipids with one or more double bond can undergo lipid peroxidation, but polyunsaturated species (more than one double bond) are more susceptible to this process. Polyunsaturated fatty acids (PUFAs) present in phospholipids (e.g., phosphatidylcholine, phosphatidylethanolamine, phosphatidylinositol, cardiolipin, etc.) are among the major targets of free radical-mediated lipid peroxidation. The biochemistry of lipid peroxidation has been reviewed in detail many times [3,22], and discussing these mechanisms is beyond the scope of the present review. It is important to note, however, that the peroxidation of lipids leads to the formation of highly reactive species, including epoxides, hydroperoxides, and aldehydes. The generation and accumulation of these compounds combined with the cellular inability to detoxify them are key factors in the ferroptotic cell death cascade. This is especially true in newborns who possess immature and incomplete antioxidant defenses as well as high concentrations of PUFAs accumulated in the brain during early life to support brain development [23,24].

Within the last decade, the association between lipid peroxidation and cell death has become more relevant, with the description of a regulated necrotic cell death modality characterized by increased phospholipid peroxidation [4,25,26,27]. The use of different small molecules and genetic animal models then led to the understanding that this method of cell death was iron-dependent [28], leading to its name: ferroptosis. Although phospholipid oxidation products are increased during the ferroptosis cascade [27], cardiolipin (a species found exclusively in the mitochondria) products are not increased. It is not clear yet whether they are not formed or they are rapidly consumed during the process [29]. There are multiple ferroptosis pathways, many of which revolve around iron, acyl-CoA, and glutathione (GSH) metabolism (particularly GSH synthesis and the activity of GPX4). These pathways are demonstrated in Figure 1, and some of the key pathways are further described below.

### 2.2. The Cystine/Glutamate Antiporter (Xc-)

GSH is among the most important small-molecule antioxidants in cells, where it is found in the range of 1–10 mM [30]. It is a tripeptide synthesized from glutamate, glycine and cysteine. The antioxidant activity of GSH relies on its ability to quench highly reactive species, including hydroxyl radicals, peroxyl radicals, and alcoxyl radicals [31]. The oxidation of GSH generates glutathione disulfide (GSSG), which consists of two glutathione molecules linked by a disulfide bond. GSH is also used in the elimination of electrophilic peroxidation products (e.g., 4-hydroxynonenal [4HNE]) in reactions catalyzed by glutathione-S-transferase enzymes (GSTs) [32,33]. GSH is a substrate for glutathione peroxidase enzymes (eight isoforms, GPX1-GPX8; see Section 2.3 for more details), which reduce peroxides to their corresponding alcohols. The GSH/GSSG ratio can be used to estimate the overall redox status of a cell [34]. As cellular oxidative stress increases, GSH is consumed and GSSG is generated, decreasing the GSH/GSSG ratio [34]. Decreases in this ratio have been described in many (patho)physiological processes, including neurodegeneration, ischemia-reperfusion injuries and during the ferroptotic cell death cascade [35,36].

The rate-limiting step in the endogenous synthesis of GSH is the uptake of cysteine to the cytosol. Cysteine is obtained by the cells via the cystine/glutamate antiporter (Xc-) system, which relies on importing cystine (the oxidized form of cysteine) at the expense of a glutamate in a 1:1 ratio [37]. Once inside the cell, cystine is rapidly reduced to cysteine and then utilized for GSH synthesis [38]. In 2012, Dixon et al. utilized erastin, an irreversible inhibitor of the Xc- system, to demonstrate that cells died by ferroptosis when deprived of their cystine uptake mechanism [28]. The cystine deprivation halted the synthesis of GSH resulting in increased lipid peroxidation and cell death [28]. The synthesis of GSH can also be inhibited by buthionine sulfoximine (BSO), sulfasalazine, glutamate, and sorafenib, and several studies have demonstrated that the administration of these compounds increases the cellular susceptibility to ferroptosis [26,39,40]. Inversely, attempts have been made to boost the synthesis of GSH by using the non-toxic precursor N-acetylcysteine (NAC), which could modulate ferroptosis and have therapeutic applications [41,42].

### 2.3. The Glutathione Peroxidase 4 (GPX4) Pathway

One of the main mechanisms to detoxify phospholipid hydroperoxides is GPX4. GPX4 is a seleno-cysteine enzyme that reduces phospholipid hydroperoxides to their corresponding hydroxides at the expense of GSH and NADPH [43]. Removing or decreasing the cellular GPX4 activity (either by genetic manipulation or chemical inhibition) leads to increased lipid peroxidation and accumulation of phospholipid hydroperoxides [44]. Selectively deleting GPX4 in neurons results in mortality in the neonatal period [45,46], while conditionally deleting this enzyme in adult animals leads to neurodegeneration, mitochondrial oxidative damage and ferroptosis [47]. Similarly, the use of RSL3, a specific inhibitor of GPX4, leads to increased lipid peroxidation and associated ferroptosis [48,49]. It is worth mentioning that the inhibition of GSH synthesis described in the previous section also affects the GPX4 activity, as GSH is a critical co-substrate that allows for the reduction of phospholipid hydroperoxides [3]. Hence, compromising the cysteine-GSH-GPX4 system leads to the accumulation of lipid peroxidation products and associated ferroptotic cell death.

### 2.4. The Roles of ACSL4 and LPCAT3 in Ferroptosis Sensitivity

The enzyme acyl-CoA synthetase long-chain family member 4 (ACSL4) also plays a central role in the regulation of ferroptosis. ACSL4 is required for activation of ω-6 PUFAs, especially arachidonic acid (AA) and adrenic acid (AdA) to their acyl-CoA derivatives, allowing for their incorporation into phospholipids [50]. AA and AdA are highly oxidizable fatty acids, and in a scenario where lipid peroxidation is increased, they become primary targets for oxidation [27]. In 2016, Doll et al. demonstrated that modulating ACSL4′s activity affects the cellular sensitivity to ferroptosis [51]. As described in Section 2.3, knocking out GPX4 in cells led to increased lipid peroxidation and ferroptosis. The introduction of a double knockout (both ACSL4 and GPX4) in the cells translated into an increased resistance to ferroptosis [51]. Since ACSL4 allows for the esterification of highly oxidizable PUFAs for incorporation into the cell membrane, its knockout depletes the membrane of PUFAs, resulting in a less oxidizable membrane and decreased phospholipid peroxidation and ferroptosis. These findings suggest that manipulating ACSL4 activity can be a potential therapeutic approach for conditions where ferroptosis is a contributing factor (e.g., HIBI). Notably, increasing the levels of monosaturated fatty acids (MUFAs) translates into increased resistance to ferroptosis [52]. The supplementation with MUFAs changed the cellular lipidome, decreasing the incorporation of PUFAs in the membrane and, therefore, suppressing ferroptosis.

The incorporation of PUFAs in the cell membrane requires two steps. ACSL4 first activates the fatty acid by adding a CoA unit, as described above. The next step is catalyzed by lysophosphatidylcholine acyltransferase 3 (LPCAT3), an enzyme that takes the PUFA-CoA (AA-CoA and AdA-CoA) and synthesizes the corresponding phospholipid-PUFA [27,53]. LPCAT3 catalyzes such reactions with phosphatidylcholine (PC) and phosphatidylethanolamine (PE). It has been shown that modulating LPCAT3 activity affects resistance to ferroptosis [27]. Suppression of LPCAT3 activity translates into increase resistance to ferroptosis, which can be attributed to the lower abundance of highly oxidizable PUFAs in the membrane; less oxidizable substrate translates to less phospholipid peroxidation and suppression of ferroptosis.

### 2.5. The Role of Hypoxia-Inducible Factors (HIFs) in Ferroptosis Sensitivity

HIF1α is the 3′ enhancer of the erythropoietin gene and contains three domains: the N-terminal domain, the C-terminal transactivation domain which primarily regulates gene transcription activity, and an oxygen-dependent degradation domain which mediates oxygen-dependent stability [54]. HIF1α suppresses ACSL4 after adult ischemic stroke, thereby attenuating neuronal death and inhibiting microglial cytokine production [55]. HIF1α has also been shown to promote intracellular lipid storage in lipid droplets, reducing fatty acid oxidation and protecting cells from ferroptosis [56]. Though less studied than the 1α isoform, HIF2α also likely plays an inverse but similarly important role in ferroptosis. HIF2α has been shown to selectively enrich lipids containing polyunsaturated fatty acyl side chains, resulting in a more ferroptosis-sensitive cell state [57].

The HIF-related EGLNs are oxygen- and iron-dependent enzymes that may also alter cellular ferroptosis. EGLNs are proline hydroxylases (PHDs) that hydroxylate proline residues on HIF1α, resulting in increased HIF1α degradation and increased susceptibility to ferroptosis [58]. EGLN knockdown by hypoxia or chemical inhibition results in increased HIF1α expression [56] and increased ferroptosis resistance.

### 2.6. FSP1-CoQ_10_-NADPH Pathway

Ferroptosis suppressor protein 1 (FSP1)—previously known as apoptosis-inducing factor mitochondria-2 (AIFM2)—is a key GPX4-independent modulator of lipid peroxidation. FSP1 catalyzes the NAD(P)H-dependent reduction of coenzyme Q_10_ (CoQ_10_, also known as ubiquinone) to ubiquinol (CoQ10H_2_), which is a potent antioxidant that attenuates lipid peroxidation [59,60,61]. CoQ_10_ is a lipophilic molecule present in every cell membrane, especially in the inner mitochondrial membrane, where it is responsible for transporting electrons from complexes I and II to complex III [62,63]. CoQ_10_ deficiency has been linked to many diseases, including encephalomyopathy, cerebellar ataxia, ischemic heart disease, and metabolic syndrome [64,65].

In addition to its role in oxidative phosphorylation, CoQ_10_ is a potent antioxidant [64,66,67,68]. CoQ_10_ rapidly reacts with alcoxyl and peroxyl radicals, thus preventing the propagation of lipid peroxidation [69]. Studies have shown that deficiency in CoQ_10_ leads to increased lipid peroxidation markers, including malondialdehyde (MDA) [70]. More recently, it has been shown that altering CoQ_10_ metabolism modulates ferroptosis [71].

### 2.7. The Role of Lipoxygenases in Ferroptosis

Lipid peroxidation occurs via two main mechanisms: metal-mediated peroxyl radical formation and an enzyme-mediated pathway catalyzed by lipoxygenases (LOXs) [72,73,74]. Two specific LOXs have been associated with ferroptotic cell death: 15-lipoxygenase (15-LOX-1) and 12-lipoxygenase (12-LOX) [72,73,74]. In 2017, Wenzel and co-workers described the pro-ferroptotic effects of 15-LOX-1. The authors used phosphatidylethanolamine-binding protein 1 (PEB1), a protein inhibitor that complexes with 15-LOX and decreases its activity, to demonstrate that the inhibition of 15-LOX-1 suppressed ferroptosis [74]. Moreover, while the pharmacological inhibition of 15-LOX-1 increased resistance to ferroptosis in *gpx4^−/−^* mice [75], the genetic manipulation of *alox15* (gene that encodes 15-LOX-1) did not affect the course of ferroptosis in *gpx4^−/−^* mice [76]. More recently, Shah and their co-workers utilized cell culture models to characterize the contributions of 5-LOX, 12-LOX and 15-LOX-1 [73]. The overexpression of the three LOX isoforms increased the sensitivity of cells to ferroptosis. Notably, LOX-mediated ferroptosis was suppressed by liproxstatin-1 and other antioxidants, indicating that progression of the lipid peroxidation cascade is vital for cell death [73]. Pharmacological inhibitors of 5-LOX failed to protect cells from ferroptosis, however, suggesting that not all LOX isoforms contribute equally to the process [73]. In their conclusion, the authors proposed that LOXs contribute to the overall pool of hydroperoxides, but the main contributor to the ferroptotic cell death is the uncontrolled progression of lipid peroxidation.

## 3. Mechanisms and Therapeutic Targeting of Ferroptosis in HIBI

Many of the ferroptosis pathways have been demonstrated to be altered after neonatal HIBI, and most of them have at least one published study demonstrating an attempt to alter the pathway for neuroprotection (Figure 2, Table 1).

### 3.1. Oxidative Stress and Lipid Peroxidation

#### 3.1.1. Lipid Peroxidation and HIBI

It is widely accepted that oxidative stress is a key component of neonatal HIBI; a concept that is supported by studies demonstrating alterations in markers of oxidative stress such as MDA and GSH. Blood MDA concentrations have been shown to be increased in cord blood and at 48 h of life in infants with perinatal asphyxia compared to healthy controls [85]. Attenuation of oxidative stress is thought to be one of the mechanisms of action of therapeutic hypothermia, as 72 h of hypothermia reduced MDA levels compared to normothermic controls [86].

MDA and GSH are nonspecific biomarkers for cell injury and protection, however, and other assays are more specific for the lipid peroxidation that occurs during ferroptosis compared to other etiologies of oxidative stress. For instance, one study has assessed lipid peroxidation urinary biomarkers in the first 6 h of life and at 12, 24, 48, 72, and 96 h after initiation of hypothermia [87]. The investigators demonstrated that urine isoprostanes increased over time and prostaglandins decreased. Similarly, lipid peroxidation (LPO) concentrations measured by ELISA may correlate with severity of encephalopathy, Apgar score, and initial blood pH. Additionally, LPO concentrations were higher in infants with asphyxia who died compared to those who survived [88]. LPO elevation had 89% sensitivity, 96% specificity, 96% PPV, and 90% NPV for the diagnosis of neonatal hypoxic-ischemic encephalopathy [89].

#### 3.1.2. Interventions Targeting Lipid Peroxidation after HIBI

Several current and potential interventions non-specifically affect or specifically target oxidative stress and lipid peroxidation. One of the mechanisms by which therapeutic hypothermia—the only proven effective therapy in neonatal HIBI—is thought to confer neuroprotection is in part through reduction of free radical production [90,91]. Other investigative therapies include metformin, which suppresses mitochondrial complex I, thereby restricting mitochondrial respiration [92].

Other scavengers of oxygen free radicals and general antioxidants that have been tested in neonatal HIBI include allopurinol [77], melatonin [83], and N-acetylcysteine [93,94]. Vitamin D has also been used with and without NAC to improve the cellular redox state after oxidative brain injury [95,96,97]. Similarly, vitamin E (α-tocopherol) is a well-studied antioxidant that when administered prior to injury in a piglet model of hypoxic brain injury may decrease lipid peroxidation [98]. The antioxidant action of vitamin E is supported further by administration of vitamin C, which facilitates the return of the harmful oxidized α-tocopherol radical back to α-tocopherol [99,100]. In addition to several other potential mechanisms of action in neonatal HIBI, erythropoietin may also exert its neuroprotective effects on lipid peroxidation [101,102].

Although lipid peroxidation is a key mechanism of ferroptotic cell death, it is not specific to ferroptosis alone. The remainder of this review will focus on the data surrounding the involvement of ferroptosis-specific pathways in neonatal HIBI pathophysiology.

### 3.2. Fenton Reaction

#### 3.2.1. Fenton Reaction and HIBI

The iron-based Fenton reaction is one of the extrinsic or transporter-dependent ferroptosis pathways that has been associated with neonatal HIBI. Newborns with severe asphyxia have been shown to have significantly elevated levels of free (not protein-bound) iron and lipid peroxidation in the plasma throughout the first 24 h of life. Additionally, those infants with severe asphyxia that went on to have abnormal neurodevelopment at 1 year of age were also more likely to have detectable levels of free iron in the plasma compared to those with normal development [103]. This was subsequently confirmed in the newborn lamb model of HIBI. In the animal model, the elevation in plasma free iron was also associated with increased MDA concentrations and decreased alpha-tocopherol and ascorbic acid/dehydroascorbic acid concentrations [104]. Neither of the previous studies assessed brain concentrations of iron products, but preliminary investigations in brain tissue have been performed in the rat model. At 72 h after neonatal HIBI in the rat model, the ipsilateral brain demonstrated significantly increased expression of transferrin receptor, ferritin heavy chain, and ferritin light chain [9]. It is important to note, however, that there is a balance that must be achieved with regards to iron concentrations in the brain, as too little iron can also be injurious and also leads to adverse neurodevelopmental outcomes [105].

#### 3.2.2. Interventions Targeting Fenton Reaction after HIBI

Investigations into targeting this pathway have included attempts to decrease free iron production or scavenge free iron products to prevent the production of free radicals, or to attenuate the injury caused by iron radicals that have already been formed. One study used N-omega-nitro-L-arginine to inhibit nitric oxide formation and showed that administration immediately after HIBI resulted in significantly lower plasma concentrations of free iron as well as lower MDA [104]. Iron chelators such as deferoxamine have also been used in multiple studies of neonatal HIBI to attempt to decrease free iron products in the blood and brain [106,107,108,109,110], though no human studies have yet been reported. Similarly, administration of exogenous erythropoietin may inhibit Fenton chemistry, potentially through the iron scavenging process of erythropoiesis [80,81]. Lastly, vitamin D—specifically 1,25-hydroxyvitamin D—also has the potential to decrease iron-induced lipid peroxidation and neuronal injury [111,112].

### 3.3. GPX4 and System Xc-

#### 3.3.1. GPX4 and System Xc- Pathway

GPX4 activity is a key regulator of ferroptosis and has been shown to play an important role in neonatal HIBI pathophysiology. Although the cumulative level of all GPX proteins does not appear to be altered after neonatal HIBI in mice [113], the GPX4 isoform protein concentration is decreased [10] along with decreases in GSH and 4-HNE in the ipsilateral hippocampus after HIBI compared to sham controls [84]. GPX concentrations were found to be elevated in the cerebrospinal fluid of neonates with HIE, correlating with encephalopathy severity (measured by modified Sarnat staging), arterial cord blood pH, and neurodevelopment at one year as measured by the Denver Developmental Screening Test [114]. These results are complicated somewhat by the inclusion of preterm infants (as low as 32 weeks of gestation) in the study and the lack of specific GPX4 measurements, however.

GPX overexpression in a transgenic mouse model resulted in significantly decreased histologic brain injury scores after neonatal HIBI [113,115]. This upregulation of GPX results in increased FLICE inhibitory protein (FLIP) which is a key regulator in the Fas death receptor pathway and inhibits cleavage of caspase-8 into its active form [116]. The endogenous upregulation of GPX after hypoxic-preconditioning may be in part responsible for the neuroprotective effects of the preconditioning against HIBI, an effect that was not replicated by increasing GSH concentrations alone [115]. Although GPX4 has been the most thoroughly studied GPX isoform, GPX1 knockout was also shown to increase injury after ischemia/reperfusion in adult brains [117,118].

For the System Xc-, hypoxia alone results in increased expression of SLC3A2 immediately after injury in the forebrain of a mouse model [119]. This increase in system Xc- activity and the resulting increase in cystine uptake has also been associated with hypoxic preconditioning in neural stem cells [120]. Conversely, after HIBI, decreased SLC7A11 protein expression is seen in the ipsilateral hemisphere at both 24 and 72 h after injury and is associated with decreased GSH and GPX4 [9,10]. The difference between the increased SLC3A2 in the first study and the decreased SLC7A11 in the latter study could be due to the different pathophysiology in hypoxia alone versus hypoxia-ischemia, but also could be related to the timing of evaluation given the multiphasic nature of hypoxic-ischemic injury.

#### 3.3.2. Interventions Targeting GPX4 Pathway after HIBI

Several interventions have been shown to target or directly alter GPX4 signaling. For example, targeted manipulation of GPX4 may be possible through the use of microRNA antagonism (resulting in increased GPX4 expression), as was demonstrated in renal ischemia and reperfusion where investigators demonstrated decreased renal injury with silencing of miR-182-5p and miR-378a-3p [121]. Intraperitoneal injection of adaptaquin, which is a hypoxia-inducible factor prolyl 4-hydrolase (HIF-PHD)-inhibitor, to the neonatal mouse model of HIBI also resulted in increased GPX4 mRNA expression in the cortex. This increase was associated with decreased cell death in the cortex and overall reduction in infarct volume, though the tissue-sparing effect was more pronounced in males than females [122].

A less targeted but more clinically feasible approach at this time would be to use an FDA-approved medication to alter GPX4 expression. For instance, melatonin administration after neonatal HIBI attenuated GPX4, GSH, and 4-HNE changes, with subsequent improvement in behavioral outcomes [84]. Erythropoietin also may be able to increase GPX expression. The only study to date to address this question, however, measured general GPX expression rather than the specific sub-types [82]. This is an important distinction because although multiple members of the GPX family can detoxify H_2_O_2_ and fatty acid hydroperoxides, GPX1 and others are not able to reduce hydroperoxides in membrane lipids to the extent of GPX4 [123]. Further studies will be necessary to confirm the specific mechanism by which erythropoietin attenuates lipid peroxidation.

### 3.4. FSP1-CoQ10-NADPH Pathway

Knockout of Aifm2/Fsp1 resulted in greater brain injury at 72 h after HIBI, associated with increased expression level of AIF1 [124]. Additionally, CoQ_10_ supplementation has shown some promise as a neuroprotective intervention in models of adult brain ischemia [125,126]. Although CoQ_10_ administration has not yet been explored in neonates, the mitochondria-targeted ubiquinone compound mitoquinol has been trialed in the rat model of neonatal HIBI. The group only assessed the number of medium-spiny neurons in the striatum, however, and found no difference between mitoquinol and vehicle control [127].

One key factor in the modulation of CoQ_10_ homeostasis relies on the manipulation of its biosynthesis. CoQ_10_ is synthesized de novo using isoprene units as building blocks [128,129]. The isoprene units are provided by the mevalonate pathway, and therefore can be manipulated by the use of statins [128,130,131]. Indeed, a meta-analysis study shows that statin treatment decreased circulating CoQ_10_ levels [132], and other studies suggest that CoQ_10_ supplementation can increase its circulating levels in patients taking statins [131,133]. All together, these observations suggest that ferroptosis can be modulated by statins and that many of the statin-associated myopathies could be related to lipid peroxidation and associated ferroptosis, which should be the topic of future investigations.

### 3.5. Downstream Pathways

The role of TLR4 in ferroptosis is complex. Although ferroptotic regulation of 4-HNE and oxidized phospholipids results in activation of TLR4 and subsequent induction of proinflammatory cytokines through NF-κB activation [134], it has also been hypothesized that neonatal HIBI increases ferroptosis by direct activation of TLR4. To support this hypothesis, Zhu, et al. demonstrated that HIBI increased TLR4 expression, and inhibition of TLR4 using TAK-242 resulted in decreased hippocampal expression of ferroptosis-related genes ATP5G3, PTGS2, CS, IREB2, and RPL8, and increased SLC7A11 and GPX4 in both cell culture and rat models of neonatal HIBI [10]. These data suggest that, at least in neonatal HIBI, TLR4 may also provide upstream regulation of ferroptosis in addition to being upregulated downstream from ferroptosis.

The ferroptosis-related release of danger-associated molecular patterns (DAMPs) such as high-mobility group box 1 (HMGB1) and the reactive aldehydes 4-hydroxynonenal (4-HNE) and MDA has been used to develop biomarkers of ferroptotic injury. These chemicals are also in themselves pro-inflammatory, however, so may additionally provide targets for therapeutic intervention. For example, HMGB1 is a transcription factor involved in chromatin remodeling and inflammation signaling. HMGB1 is released from injured cells and contributes to brain injury after neonatal HIBI in part through microglial regulation [135,136]. It was previously shown in cancer cells that knock down HMGB1 decreased erastin-induced ferroptosis through the RAS-JNK/p38 pathway [137], and a very recent study demonstrated that HMGB1 inhibition by glycyrrhizin administration decreased neuronal ferroptosis in both a cell culture oxygen-glucose deprivation model and after neonatal HIBI in rats in vivo [138].

## 4. Conclusions

Ferroptotic cell death can result from the activation of multiple pathways that result in uncontrolled lipid peroxidation. Recent studies have suggested that ferroptosis occurs after neonatal HIBI and that attenuating ferroptosis-related changes after injury may be neuroprotective. Despite the promising early studies, several significant questions remain unexamined in the current literature. These include outstanding questions regarding several of the specific mechanisms of ferroptosis, such as the role of other ferroptosis-associated pathways that have not yet been demonstrated in HIBI (such as KRAS [139]) and the role of stress granule formation [140]. Additional HIBI-specific questions include the temporal nature of the cellular changes after HIBI and whether ferroptosis follows the timing of the three canonical phases of injury. Lastly, many unanswered questions remain surrounding the use of the interventions listed in Figure 2 (and others not yet investigated), including whether the suppression of ferroptotic cell death simply redirects the injury to one or more of the other cell death pathways known to affect cells after neonatal HIBI. Ultimately, however, the current literature suggests that there is considerable promise that attenuating ferroptosis could decrease brain injury and improve outcomes in this high-risk population.

## Figures and Tables

**Figure 1 ijms-23-07420-f001:**
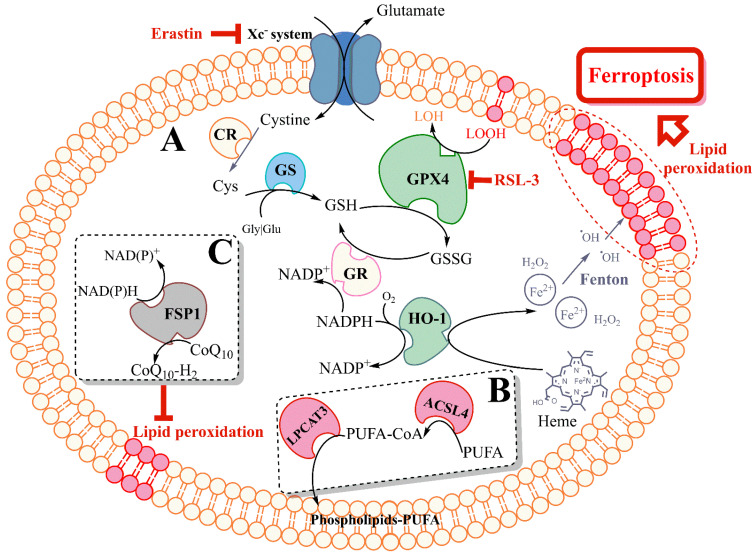
Pathways involved in ferroptotic cell death. (**A**) The cysteine-GSH-GPX4 system plays a central role in the cellular antioxidant mechanism and ferroptosis prevention. Inhibition of the Xc- system by erastin deprives the cells of cysteine, thus depriving them of GSH, triggering ferroptosis. Inhibition of GPX4 by RSL3 deprives the cells from the enzyme that reduces phospholipid peroxides (LOOH) to their corresponding alcohols (LOH), increasing lipid peroxidation and associated ferroptosis. Both erastin and RSL3 have been extensively used to induce ferroptosis in cell culture models. (**B**) The enzyme acyl-CoA synthetase long-chain family member 4 (ACSL4) activates highly oxidizable PUFAs, which is a critical step in their incorporation into phospholipids and into the cell membrane. Depleting the cells of ACSL4 increases the cellular resistance to ferroptosis. LPCAT3 catalyzes the incorporation of PUFA-CoA into phospholipids. Suppressing LPCAT3 activity increases resistance to ferroptosis. (**C**) FSP1 catalyzes the regeneration of CoQ_10_ at the expense of NAD(P)H. CoQ_10_-H_2_ (reduced form) is an effective lipid-soluble antioxidant, acting by directly reacting with free radical species in the membrane and preventing lipid peroxidation. CR, cystine reductase; GS, glutathione synthetase; GSH, reduced glutathione; GSSG, glutathione disulfide; GR, glutathione reductase; FSP-1, ferroptosis suppressor protein 1; CoQ_10_, coenzyme Q_10_; HO-1, heme oxygenase 1; LPCAT3, lysophosphatidylcholine acyltransferase 3; PUFA, polyunsaturated fatty acid.

**Figure 2 ijms-23-07420-f002:**
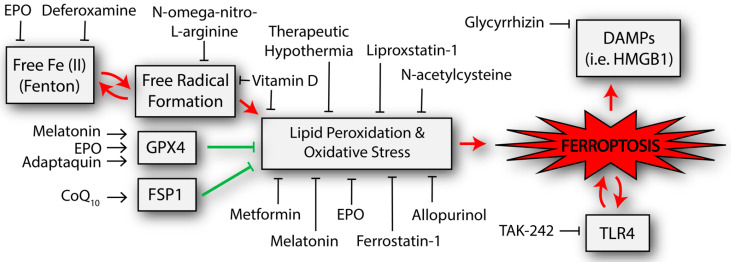
Potential neuroprotective interventions that have been shown to affect the ferroptosis-related pathways in studies of neonatal hypoxic-ischemic brain injury. Other interventions that have been trialed in adult populations but not yet neonates are described in the text but not included in the figure. The double arrows represent byproducts that can further potentiate the initial process (e.g., TLR4 is increased from the process of ferroptosis but TLR4 appears to also activate ferroptosis). CoQ_10_, coenzyme Q_10_; DAMPs, damage associated molecular patterns; EPO, erythropoietin; FSP1, ferroptosis suppressor protein 1; GPX4, glutathione peroxidase 4; HMGB1, high mobility group box 1; TLR4, toll-like receptor 4.

**Table 1 ijms-23-07420-t001:** Ongoing clinical trials (or completed trials without published results) assessing interventions that may affect ferroptosis in neonatal hypoxic-ischemic brain injury. List reflects studies registered with Clinicaltrials.gov as of June 2022.

Intervention	Potential Mechanism	Identifier *	Status
Allopurinol	Scavenges free radicals [77]	NCT03162653	Recruiting
Caffeine	Scavenges free radicals [78]	NCT03913221	Recruiting
		NCT05295784	Not yet recruiting
Citicoline	Alters phosphatidylcholine synthesis, resulting in attenuation of lipid peroxidation [79]	NCT03181646	Unknown
Darbepoetin/ Erythropoietin	Inhibits Fenton chemistry, scavenges iron through erythropoiesis [80,81], increases GPX expression [82]	NCT00719407	Completed
		NCT03071861	Recruiting
		NCT03079167	Active
		NCT03163589	Unknown
Melatonin	Scavenges free radicals [83], attenuates GPX4, GSH, and 4-HNE changes [84]	NCT02621944	Recruiting
		NCT03806816	Recruiting

* Clinicaltrials.gov identifier.

## Data Availability

Not applicable.

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
