# Peer review of "Ferroptosis: A Promising Therapeutic Target for Neonatal Hypoxic-Ischemic Brain Injury"

_ijms, 2022, doi:10.3390/ijms23137420_

Round 1
Reviewer 1 Report
Suggestions for Authors
In this paper, Eric S. Peeples and colleagues review the current knowledge and the emerging evidence linking ferroptosis to neonatal hypoxic-ischemic brain injury (HIBI). As a new type of programmed cell death, ferroptosis has become a new therapeutic target and has gradually attracted clinical attention. Therefore, it is very meaningful to focus on the potential role of ferroptosis in HIBI. Nonetheless, many problems remain in this manuscript. To improve the quality of this manuscript I would suggest the following things:
1.Few references could be added in several places where authors have some important statements, for example:
“Ferroptosis is a relatively recently described mechanism of cell death.” (line 29)
“Although oxygen-glucose deprivation …… several parallel cell death mechanisms.” (line 63-65)
“This is especially true…… and incomplete antioxidant defenses.” (line 86-87)
“It is worth mentioning that…… for the reduction of phospholipid hydroperoxides.” (line 155-157)
Also, I wonder why these articles are not cited: (PMID: 25006720, PMID: 31710357).
2. Authors did not describe the role of another membrane remodeling enzyme-LPCAT3 in “2.4. The Role of ACSL4 in Ferroptosis Sensitivity” (line 160). LPCAT3 is also an important factor in ferroptosis. The ligation of long-chain PUFAs and coenzyme A catalyzed by ACSL4 requires re-esterification to phospholipids by various LPCAT enzymes, thereby increasing the incorporation of long-chain PUFAs into lipid and membrane structures in cells. The author need to supplement relevant descriptions to correspond to Fig1.
3. Similar to the mechanism above, I think the description “increasing the levels of other ACSL isoforms (e.g. ACSL3) does not affect the cellular susceptibility to ferroptosis” is not quite correct. ACSL3-dependent enrichment of membrane MUFAs has been reported to reduce cellular exposure to ferroptosis. Thus, they may want to take a look at this paper on the study of exogenous MUFAs promoting ferroptosis-resistant cell state (PMID: 30686757). Furthermore, regarding the relationship between the expression level of ACSL4 and the sensitivity of cells to ferroptosis, this phenomenon was actually observed in a subset of cells in triple-negative breast cancer as well as in drug-resistant mesenchymal cancer cells. Also, there appear to be no studies showing that regulation of ACSL4 can interfere with disease progression in non-cancer diseases, including neurological diseases. Therefore, in my opinion,targeting ACSL4 to intervene in HIBI does not seem feasible.
4. The author did not describe the enzymatic reactions proposed in ferroptosis, which should include proteins such as LOX, the first paper about it: (PMID: 29053969, 2017, Cell) and other papers published in >2017. In addition, some pharmacological inhibitors of LOX have been reported to inhibit ferroptosis, and knockout of 12/15-LOX or application of the LOX inhibitor baicalin can protect mice from ischemic brain injury.
5. In “3.1.1. Lipid Peroxidation and HIBI”, the authors should point out the characteristics of the neonatal brain, such as being rich in PUFAs, low in antioxidant ability, and therefore particularly vulnerable to free radical attack. In addition, this paper didn’t include some recent important findings. For example, study has shown that gastrodia pretreatment can protect against glutamate-induced ferroptosis in neuronal cells (PMID: 31698019). And antioxidant carvacrol can reduce the level of lipid peroxides in ischemic brain tissue and inhibit the occurrence of ferroptosis in hippocampal neurons, thereby improving the brain damage caused by ischemia (PMID: 31470002).
6. Several cell death forms, including apoptosis, necrosis, autophagy and pyroptosis, have been shown to be closely related to neonatal HIBI. And when upstream damaging response is greater than the protective response, if one form of cell death is inhibited, another death pathway may intervene. The authors should consider discussing interventions targeting ferroptosis in this case.
7. Line 212 has grammatical problem and line 388 has reference format error.
Reviewer 2 Report
A timely review article by Dr. Peeples and Dr. Mattos elaborates on the role of ferroptosis in Neonatal Hypoxic-Ischemic Brain Injury and discusses its therapeutic intervention. It's a very well-written review article with proper elaborative figures. A few things need to be addressed before it's ready for acceptance. They are as follows:
1. Under the downstream pathways subsection where authors have mentioned that "cancer cells that knockdown of HMGB1 384 decreased erastin-induced ferroptosis through the RAS-JNK/p38 pathway" authors should add a few lines on how downstream of GPX4 oncogenic pathways will be impactful. To add a few lines on this aspect- authors might add a few lines on the on the following topic.
It has been discussed recently how KRAS might play some role in ferroptosis and lipid biogenesis (PMCID: PMC8045781), while it has been shown recently that mutant KRAS activates NRF2 antioxidant pathways and NRF2 activates glutaminolysis and glutamine deprivation caused GPX4 level reduction. It is an interesting aspect that NRF2 plays a significant role in lipid peroxidation and ferroptosis (PMCID: PMC7185043 and PMCID: PMC6859567). This aspect of the role of glutamine metabolism in regulating GPX4 might be one of the future areas of research in the field. It will be worthwhile to add a few lines on this aspect. Also, it has been shown that NRF2 plays a role in regulating brain injury (PMID: 32269897).
2. Authors should add a table mentioning the drug names which are in the clinical trial stage targeting ferroptosis in neonatal brain injury.
3. It will be worthwhile if a few lines would be added to the historical timeline of ferroptosis discovery.
4. It has been shown that traumatic brain injury can induce stress granule formation PMID: 29432563. It has been shown (PMCID: PMC8073197) stress granules might play some role in ferroptosis There might be some possibility that the interconnection between ferroptosis and stress granules might have some implication in brain injury therapeutics. Authors should add a few lines on this as one of the future aspects of the field.
Round 2
Reviewer 1 Report
THANKS FOR THE REPLY.
Reviewer 2 Report
All concerns have been addressed, ready for acceptance.